

**Evaluation of Single- and Multiple-Doppler Lidar Techniques to Measure**
**Complex Flow during the XPIA Field Campaign**
by
Aditya Choukulkar[1,2*], Alan Brewer[2], Scott P. Sandberg[2], Ann Weickmann[1,2], Timothy A.
Bonin[1,2], R. Michael Hardesty[1,2], Julie K. Lundquist[3,4], Ruben Delgado[5], G. Valerio Iungo[6], Ryan
Ashton[6], Mithu Debnath[6], Laura Bianco[1,7], James M. Wilczak[7], Steven Oncley[8], Daniel Wolfe[7]
*[1]Cooperative Institute for Research in Environmental Sciences, Boulder CO*
*[2]Chemical Sciences Division, National Oceanic and Atmospheric Administration, Boulder CO*
*[3]University of Colorado Boulder, CO*
*[4]National Renewable Energy Laboratory, Golden CO*
*[5]University of Maryland Baltimore County, MD*
*[6]University of Texas at Dallas, Richardson TX*
*[7]Physical Sciences Division, National Oceanic and Atmospheric Administration, Boulder CO*
*[8]National Center for Atmospheric Research, Boulder CO*
**For Submission to:**        **Atmospheric Measurement Techniques**
Corresponding Author*:        Aditya Choukulkar

20                Cooperative Institute for Research in Environmental Sciences

21                University of Colorado,

22                Boulder, CO  80305

24                Telephone:  303-497-4016

Email:  aditya.choukulkar@noaa.gov

Fax:  303-497-5318



**Abstract:**

Accurate three-dimensional information of wind flow fields can be an important tool in

not only visualizing complex flow, but also understanding the underlying physical processes and
improving flow modeling.  However, a thorough analysis of the measurement uncertainties is
required to properly interpret results.  The XPIA (eXperimental Planetary boundary layer
Instrumentation Assessment) field campaign conducted at the Boulder Atmospheric Observatory
(BAO) in Erie, CO from 2 March – 31 May 2015 brought together a large suite of in-situ and
remote sensing measurement platforms to evaluate complex flow measurement strategies.

In this paper, measurement uncertainties for different single and multi-Doppler strategies

are investigated.  The tradeoffs (such as time/space resolution vs. spatial coverage) among the
different measurement techniques are evaluated using co-located measurements made near the
BAO tower.  Sensitivity of the single/multi Doppler measurement uncertainties to averaging
period are investigated using the sonic anemometers installed on the BAO tower as the standard
reference.  Finally, the radiometer measurements are used to partition the measurement periods
as a function of atmospheric stability to determine their effect on measurement uncertainty.

It was found that with increase in spatial coverage and measurement complexity, the

uncertainty in the wind measurement also increased.  For multi-Doppler techniques, the increase
in uncertainty for temporally uncoordinated measurements is possibly due to requiring additional
assumptions of stationarity and/or horizontal homogeneity.  It was also found that wind speed
measurement uncertainty was lower during stable conditions compared to unstable conditions.











## 1. Introduction

Scanning coherent Doppler Light Detection and Ranging (LIDAR) systems have proved to be invaluable tools for wind measurements in research as well as commercial applications. A valuable advantage of scanning Doppler lidar systems is its ability to make measurements over horizontal and vertical extents using a combination of azimuthal, Plan Position Indicator (PPI) scans and vertical plane (RHI) scans. Doppler lidars measure the projection of the wind velocity along the beam pointing direction denoted as line of sight (LOS) velocity or radial velocity given in Eq. 1.

$$V_r = u \sin\theta \cos\varphi + v \cos\theta \cos\varphi + w \sin\varphi \qquad (1)$$

where, $V_r$ is the LOS velocity, $u$, $v$, $w$ are the velocity components in the east-west direction, the north-south direction, and in the vertical respectively; $\theta$ and $\varphi$ are the azimuth and elevation angles respectively. In order to derive the 2-dimensional or 3-dimensional wind velocity requires the use of suitable measurement strategies and/or velocity retrieval algorithms.

The 2-D and 3-D wind measurements from Doppler lidars are useful in various fields of study such as boundary layer meteorology (Fernando et al., 2015; Vanderwende et al., 2015), air quality (Barlow et al., 2011; Collier et al., 2005) wind energy research (Banta et al., 2015; Käsler et al., 2010; Mikkelsen, 2014; Newsom et al., 2015) among others. The simplest techniques to derive a profile of wind speed and direction using a single Doppler lidar are the Velocity Azimuth Display (VAD) technique (Browning and Wexler, 1968) and the Doppler Beam Swinging (DBS) technique (Strauch et al., 1984). These techniques assume horizontal homogeneity of the wind in the measurement volume to estimate the profile of wind speed and direction. Other techniques such as the Velocity Volume Processing (VVP) (Waldteufel and Corbin, 1979) and the "Arc Scan" technique (Wang et al., 2015) limit the assumption of horizontal homogeneity to smaller volumes within the lidar scans or to certain azimuth ranges respectively, allowing to better preserve the spatial variability information at the expense of increased uncertainty in the wind retrieval, especially when the wind direction is perpendicular to the scan sector (Krishnamurthy et al., 2013).

A common method to make wind field measurements without assumption of spatial homogeneity is through multi-Doppler techniques. "Virtual towers" (Calhoun et al., 2006) use



multiple Doppler lidars to interrogate a common volume in space in a temporally coordinated
fashion, iterating through several height in order to create a wind profile. Several configurations
of multi-Doppler scanning have been tested to quantify the skill in deriving two and three-
dimensional wind fields. For example, co-planar RHI scans were used to study flows in
mountain valleys (Hill et al., 2010) and within a meteor crater (Cherukuru et al., 2015), co-planar
conical scans (PPIs) have been used to study coherent structures (Newsom et al., 2008; Träumner
et al., 2015) and wind turbine wakes (Vollmer et al., 2015). Three-dimensional wind field
measurements made using dual-Doppler intersecting RHI scans and using continuity to estimate
the vertical velocity were used to study flow upstream and downstream of a utility scale wind
turbine (Newsom et al., 2015). Three-dimensional wind and turbulence measurements using
fully coordinated short-range continuous wave triple lidars (Mikkelsen et al., 2008) and long-
range triple lidar scanning (Berg et al., 2015) have been demonstrated to provide high quality
measurements of complex flow. In addition, manually coordinated triple lidar measurements
(Wang et al., 2016) were also tested and showed promise in measuring the three-dimensional
wind fields operationally. The Lower Atmospheric Boundary Layer Experiment (LABLE)
validated wind and turbulence measurements from triple Doppler lidar measurements (Klein et
al., 2015; Newman et al., 2016).

In addition to multi-Doppler approaches to measuring complex flow, several techniques

enable wind field retrievals from single Doppler lidars which resolve the spatial variability
measured by the lidar. For example, the Optimal Interpolation (OI) technique allows 2-D wind
field retrievals on azimuthal scans (Choukulkar et al., 2012) without assumption of homogeneity
of the wind field. In addition, variational methods can determine the wind fields from single or
multiple Doppler lidars (Chan and Shao, 2007; Drechsel et al., 2009; Newsom et al., 2008).

The choice of the measurement strategy and the retrieval algorithms come with

assumptions inherent to their process which need to be properly understood to interpret the
measurements made. Several studies have been conducted to evaluate the measurement accuracy
of the various single and multi-Doppler techniques. For example, measurement uncertainties in
wind measurements made using the DBS technique in complex terrain were investigated by
(Bingöl et al., 2009) while (Lundquist et al., 2015) studied the uncertainties in wind
measurements using the DBS technique in presence of complex flow by simulating lidar





measurements within a wind turbine wake using a wind field created with large-eddy simulation.
Wind field measurements made using the virtual towers technique has been validated (Damian et
al., 2014; Gunter et al., 2015) to show high skill in measuring 2-D wind fields and (Stawiarski et
al., 2013) did a detailed error analysis of dual-Doppler co-planar PPI technique. Uncertainties in
three-dimensional wind field retrievals using triple Doppler lidar techniques have also been
investigated. For example, (Fuertes et al., 2014) and (Newman et al., 2016) presents a detailed
analysis of 3-D winds and turbulence measurements made using staring triple Doppler
measurements, while (Berg et al., 2015) present validation of three-dimensional wind
measurements made through continuous scanning.
While considerable effort has been devoted to evaluating each of these wind
measurement techniques, few studies have inter-compared wind measurements from multiple
Doppler lidar techniques against a common standard or discussed the trade-offs between the
different measurement techniques. The eXperimental Planetary boundary layer Instrumentation
Assessment (XPIA) field campaign conducted at the Boulder Atmospheric Observatory (BAO)
in Erie, CO from 2 March – 31 May, 2015 provided an unique opportunity to inter-compare (in
similar atmospheric conditions) various single and multi-Doppler wind measurement strategies
to measure complex flow. In this paper, precision of single and multiple Doppler lidar
techniques to measure complex flow are evaluated. In addition, the trade-offs in terms of
measurement precision, spatial coverage and temporal resolution between the various
measurement techniques are also discussed. The paper is organized as follows: the experiment
setup and measurement area is described in Section 2. Section 3 presents analysis of the LOS
velocity uncertainty and Section 4 presents results from the validation of the different
measurement techniques tested in XPIA. This is followed by a discussion of the results in
Section 5. Concluding remarks are presented in Section 6.
**2. Experiment Setup**
The XPIA field study, funded by the U.S. Department of Energy (DOE) within the
Atmosphere to electrons (A2e) program, had the goal to assess current capabilities for measuring
complex flow in and near wind farms using remote sensing instrumentation. With this goal in
mind, a large suite of instrumentation was deployed near the BAO (Kaimal and Gaynor, 1983)
facility in Erie, CO. The instrumentation included six scanning Doppler lidars (four capable of



coordinated scanning) and five vertically-profiling lidars. Lundquist et al. (2016) gives a
detailed description of the XPIA field study along with an overview of the instrumentation
deployment. Herein for sake of brevity, only the details of the scanning lidar deployment used
for testing the various single and multi-Doppler measurements are described. These lidars
included two Leosphere 200S® scanning lidars (named "D1" and "D2") and the High Resolution
Doppler Lidar (HRDL) from the National Oceanic and Atmospheric Administration (NOAA),
one Leosphere 200S® scanning Doppler lidar from the University of Texas at Dallas ("UTD")
and one Leosphere 200S® from the University of Maryland Baltimore County ("UMBC").
Figure 1 shows the deployment locations of these scanning lidars with respect to the 300-m tall
instrumented BAO tower. The pulse-width and time accumulation for each of the lidars used in
the analysis presented in this paper is given in Table 1. All the wind measurement comparisons
presented in this paper are with respect to the measurements made by the south-east sonic
anemometers installed on the BAO tower as the center of the lidar measurement volume (and the
range-gates) were always south of the BAO tower. The sonic anemometer data are filtered to
remove tower wake effects using the criteria defined in McCaffrey et al. (2016).
Table 1. Lidar operational parameters

| Lidar | Pulse Width (m) | Time Accumulation (s) |
|-------|-----------------|------------------------|
| HRDL  | 30              | 0.5                    |
| D1    | 50              | 0.5                    |
| D2    | 50              | 0.5                    |
| UTD   | 50              | 0.5                    |
| UMBC  | 50              | 1[1]                   |


---

[1] Longer accumulation time was selected to ensure sufficient range






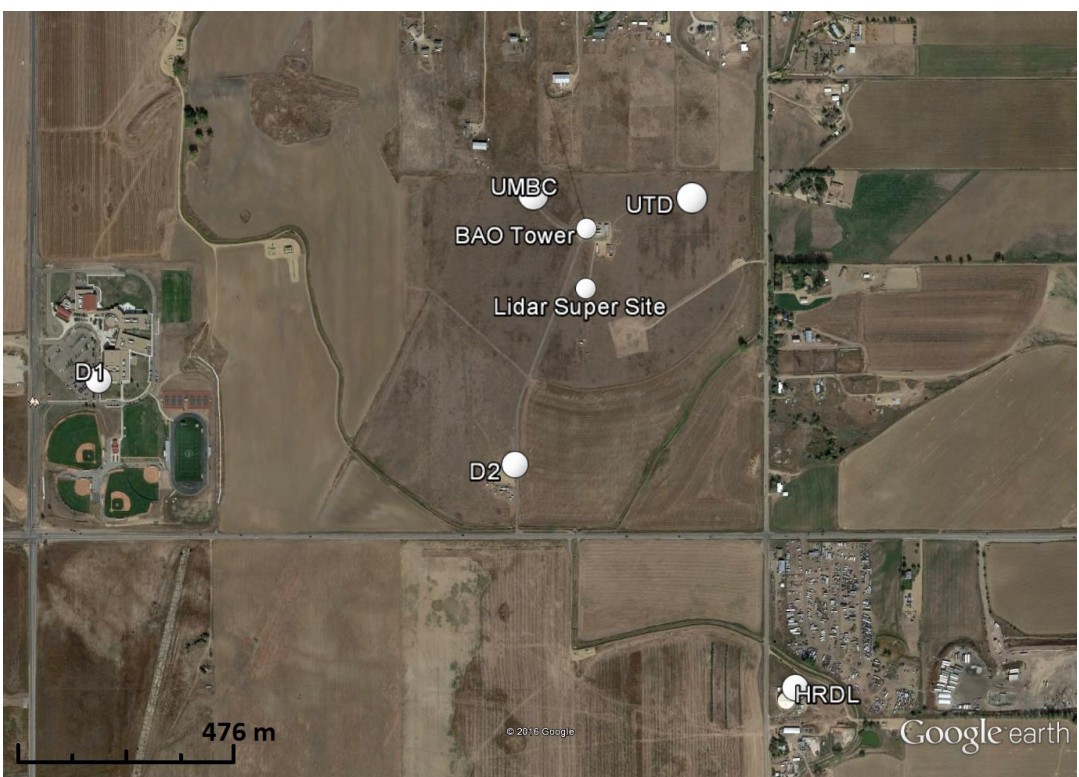

Figure 1.  Scanning Doppler lidar deployment location during the XPIA field campaign.

During the initial stages of the experiment, all the scanning Doppler lidars described

above were tested for scanner pointing accuracy.  For lidars involved in coordinated scanning
(D1, D2 and UTD), the repeatability as well as accuracy of pointing and reproducibility of time
synchronization were tested.  The details of these tests and the results are described in detail by
(Lundquist et al. 2016b) and summarized in Table 2.  The scan initialization delay estimates
from time synchronization tests for the lidars involved in coordinated scanning are summarized
in Table 3.  The scan initialization delay is defined as the time delay between the desired scan
start time and the actual scan start time.  In addition to the scan initialization delay, the delay
introduced due to each of the lidars scanning varying range of azimuths to reach the
measurement location was characterized and accounted for during the scan strategy design for
each of the measurement techniques evaluated.  The net impact of all the pointing and time
synchronization uncertainties is that all the systems could make measurements at a prescribed





location at a given time with pointing uncertainty of less than 0.15º and time uncertainty of less
than 0.4 s.
Table 2. Scanner pointing accuracy and repeatability estimates for the scanning Doppler lidars

| Lidar | Pointing Error (º) | Repeatability in AZ (º) | Repeatability in EL (º) |
|---|---|---|---|
| HRDL | < 0.1 | ~0.05 | ~0.05 |
| D1 | ~ 0.15 | 0.01 | 0.05 |
| D2 | ~ 0.15 | 0.01 | 0.05 |
| UTD | ~ 0.15 | Not Determined | Not Determined |
| UMBC | ~ 0.15 | Not Determined | Not Determined |


Table 3. Scan initialization delay estimates for the scanning Doppler lidars

| Lidar | Scan Initialization Delay (s) | Std Deviation of the Delay (s) |
|---|---|---|
| HRDL | 43.28[2] | 0.42 |
| D1 | 3.98 | 0.29 |
| D2 | 3.81 | 0.3 |
| UTD | 0.79 | 0.3 |
| UMBC | Not Applicable[3] | Not Applicable |


In this paper, the following Doppler lidar measurement techniques will be discussed:
1. Virtual tower stares
2. Coordinated triple lidar sparse sampling scans
3. Uncoordinated multi-Doppler volume scan
4. Single Doppler lidar wind retrieval using the Optimal Interpolation (OI) technique

---

[2] The unusually long delay is due to the automatic scanner calibration routine run at the beginning of each scan cycle.
[3] This system did not have ability to trigger scans at prescribed times.



*2.1 Virtual Tower Stares:*


The virtual tower stare (VTS) scan pattern involves interrogating a common volume
using multiple Doppler lidars at pre-defined heights at a given location to form a "virtual tower"
(Calhoun et al., 2006; Fuertes et al., 2014; Gunter et al., 2015; Newman et al., 2016).  A
schematic of the triple lidar VTS scan tested during the XPIA field experiment is shown in
Figure 2a.  Each of the three 200S lidars (D1, D2 and UTD) performed a temporally correlated
25-s stare at each of the 6 sonic anemometer level (50 m to 300 m with 50 m increments) and
therefore creating a virtual tower of wind measurements every 3-mins.  The LOS velocities that
fall within the common volume are least-squares fitted using Eq. 1 (Fuertes et al., 2014) to
estimate the three-dimensional wind velocity.
The common volume is defined as a square (cyan box in Figure 2b) 35 m on a side and
10 m in the vertical centered at each of the sonic height levels with its center 10 m south of the
southeast sonic anemometer on the BAO tower (this was the closest position to the tower that
allowed overlapping measurements without blockage).  As observed from Figure 2b, the
effective measurement volume (defined as the circle enveloping the outer-most range-gate
points) is slightly larger with a diameter of 60 m.  It is also seen from Figure 2b that the look
directions of D1 and UTD are close to 180 degrees apart.  This non-ideal setup for triple Doppler
measurements was dictated by logistics of deployment.  However, the UTD lidar makes
measurements with much steeper elevation angles compared to D1 and hence does provide
additional information for wind retrieval.

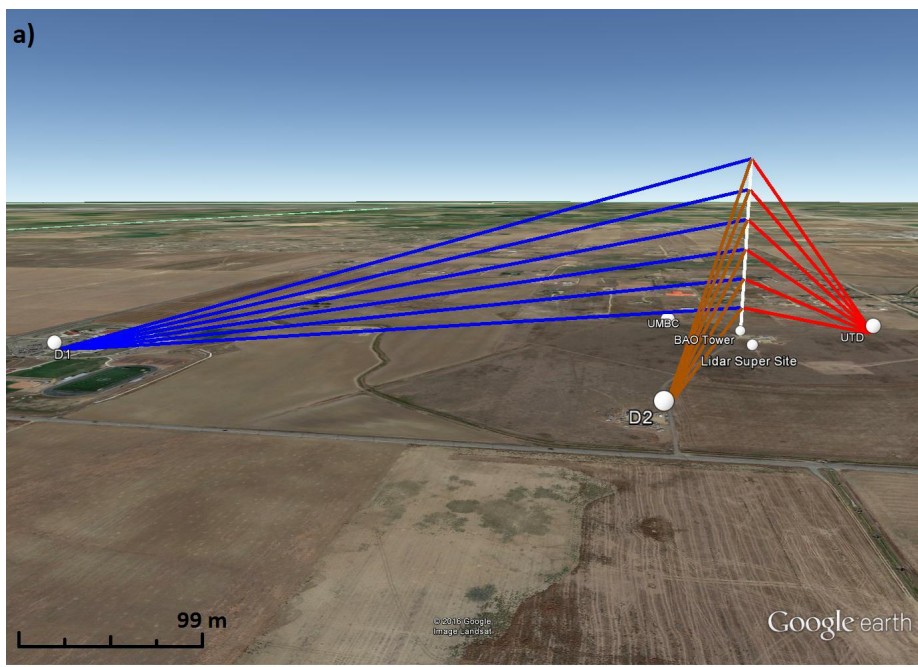

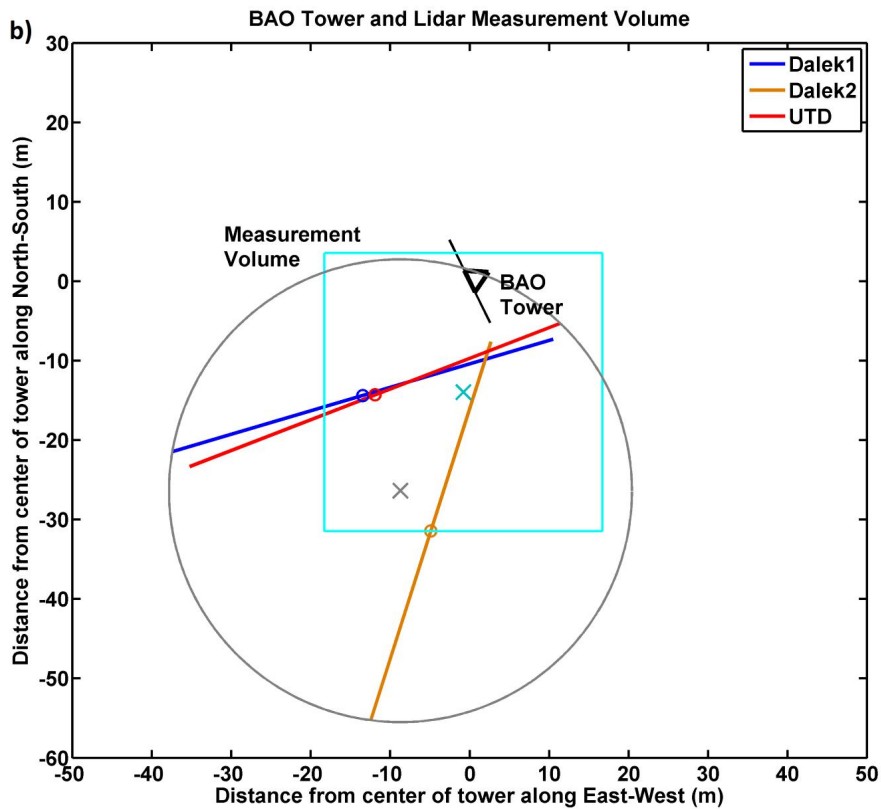





Figure 2. The Virtual Tower Stares Scan. (a) Schematic of the VTS technique. The blue, red and
orange lines indicate beams from each of the three 200S lidars that make measurements at each
sonic anemometer level. (b) Location and size of the common volume (cyan box) with respect to
the BAO tower. The blue, red and orange lines are range-gates from the three 200S lidars that
fall within the common volume. The grey circle indicates the estimated measurement volume
defined by the position of the range-gates.

Similar virtual towers were performed at two other locations to compare with other

instrumentation deployed during XPIA. Therefore the repeat period for these virtual towers
discussed here is once every 10-mins. The 25-s stare period was chosen to ensure that all three
200S lidars were measuring the common volume simultaneously. However, the 3-D wind
retrieval was made using 5-s of spatially and temporally overlapping LOS velocity data.

*2.2 Coordinated Sparse Sampling:*

While VTS scan provides a profile of wind velocity at any given location, wind velocity

measurements can also be performed over horizontal planes through temporally and spatially
coordinated scans that interrogate common volumes on a horizontal plane. One limitation to
making measurements over a large enough area using contiguous volumes is the time required to
simultaneously interrogate this area using coordinated scanning. The time required to complete a
scan is determined by the data rate of the lidar systems, overlap period and the geometry defined
by the instrument locations. Given the instrument locations during XPIA and data rate
limitations of the 200S lidars, the time required to sample an area through contiguous
measurement points would be too large to sample a feature sufficiently before it advected out of
the measurement domain. Therefore, to reduce the time required to make such a measurement,
sparse sampling strategies were considered. The sparse sampling technique discussed in this
paper is called Small Checkerboard (SCB) scan and the layout of this technique is shown in
Figure 3. The scanning strategy involved sampling a 3x3 grid covering a horizontal area of
approximately 150 m x 150 m and 100 m above the ground. The three lidars paused 5 seconds at
each grid point and hence completed one SCB scan every 1 minute. This measurement strategy
was carried out using the D1, D2 and UTD scanning 200S lidars for a period of 9 days.




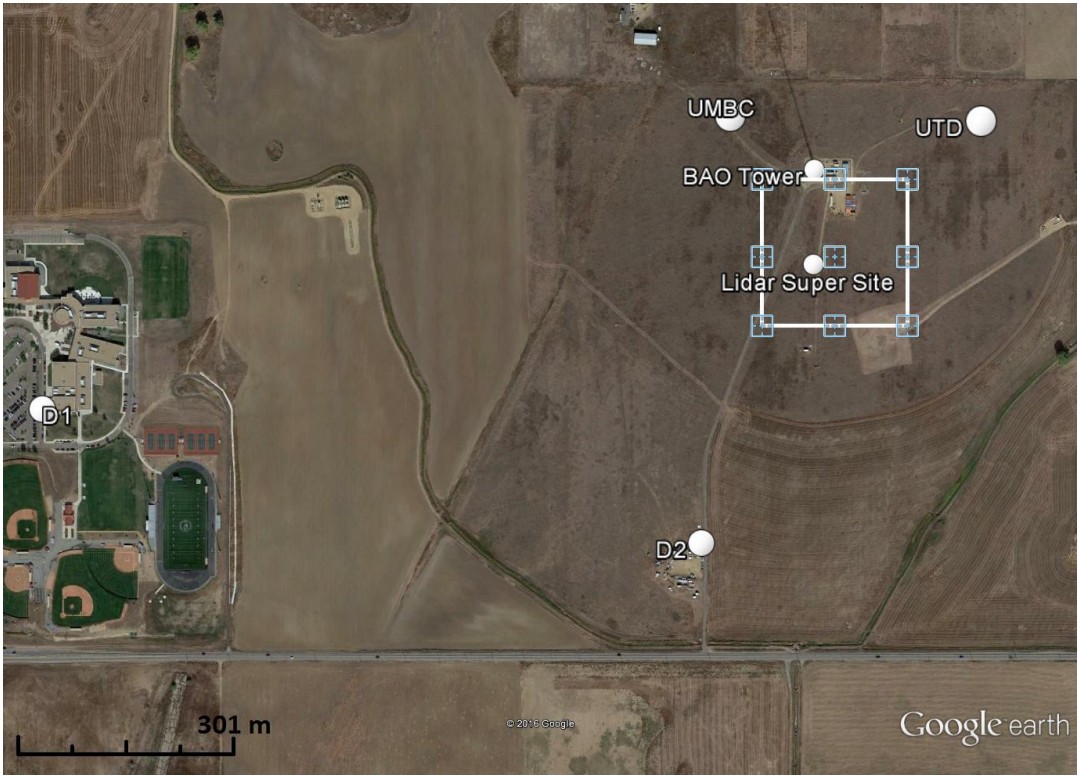


Figure 3. Schematic of the Small Checkerboard (SCB) scan. The white outline shows the

domain over which measurements are made and the blue squares indicate the locations of

measurement volumes interrogated by the scanning Doppler lidars.

*2.3 Uncoordinated Triple-Doppler Virtual Tower*

This measurement technique is similar to the one explained in Section 2.1 in that three

Doppler lidars scan a common volume to make three-dimensional wind field measurements.

However, in order to reduce the time required to perform a virtual tower and increase the update

rates, the lidars performed continuous temporally uncoordinated RHI scans at the BAO tower

location. Each RHI scan takes 15 seconds to perform and hence a three-dimensional wind field

measurement can be made every 15 seconds, compared to 3 minutes required for the VTS

technique. The trade-off is that not all lidars are looking at the same volume simultaneously. The

three dimensional wind field is estimated by least-squares fitting to Eq. 1 the LOS velocity



measurements from the three 200S lidars (D1, D2 and UTD) that fall within the common volume
(50 m on a side and 10 m in the vertical) and within a 15-s time window.  The three 200S lidars
performed intersecting RHI scans at three locations (including near the BAO tower) for 20
minutes at each location before repeating the sequence again.  This measurement strategy was
performed for a period of approximately 2 days.
*2.4 Uncoordinated Multi-Doppler Volume Scan:*

With the uncoordinated multi-Doppler measurement technique, the constraint that

multiple lidars need to interrogate a common volume simultaneously is removed allowing to
speed up sampling of the domain of interest.  In this measurement technique, five Doppler lidars
(HRDL, D1, D2, UTD and UMBC) performed a set of complementary PPI scans that would
ensure that at least two Doppler beams overlapped within each grid point (defined as 50 m on a
side in the horizontal and 15 m in the vertical) within a 5 minute update period.  This scan
strategy resulted in a grid which approximately 1.5 km x 1 km in the horizontal and covered
heights 30 m to 300 m above the ground with 15 m vertical resolution.  A representation of the
scans performed by each of the scanning lidars and the resulting grid is shown in Figure 4.







Figure 4.  Scans performed by each of the scanning Doppler lidars to produce a 3-D volume of
horizontal wind field. (a) Representation of the PPI sector scans performed by each lidar. (b)
Grid points that have LOS velocities from at least two lidars and the colors indicate difference in
azimuth between their respective look directions.

Important consideration in the scan design for this experiment were the limitations on

lidar siting and ensuring overlap with the BAO in order to validate the wind measurements.  This
constraint resulted in the measurement volume being quite close to the Doppler lidars and hence
required steep elevation angles and several PPI sectors to cover the volume of interest.  These
consideration resulted with the spatial coverage and the update rates reported in this paper.
Ideally, this type of measurement would be performed with the Doppler lidars further away from
the domain of interest so that shallower elevation PPI scans can be employed which can help
cover larger areas with faster update rates.
*2.5 Single Doppler Velocity Retrievals:*

The single Doppler retrieval technique investigated in this paper is the Optimal

Interpolation (OI) technique (Choukulkar et al., 2012).  The OI technique allows retrieval of 2-D
wind fields over PPI scans without assumption of spatial homogeneity of the wind field.  The
spatial variability information in the LOS velocity field is thus preserved which can be useful to
study complex flows such as flow in and near wind farms and in complex terrain.  An example
retrieval of the horizontal wind field using this technique is shown in Figure 5.

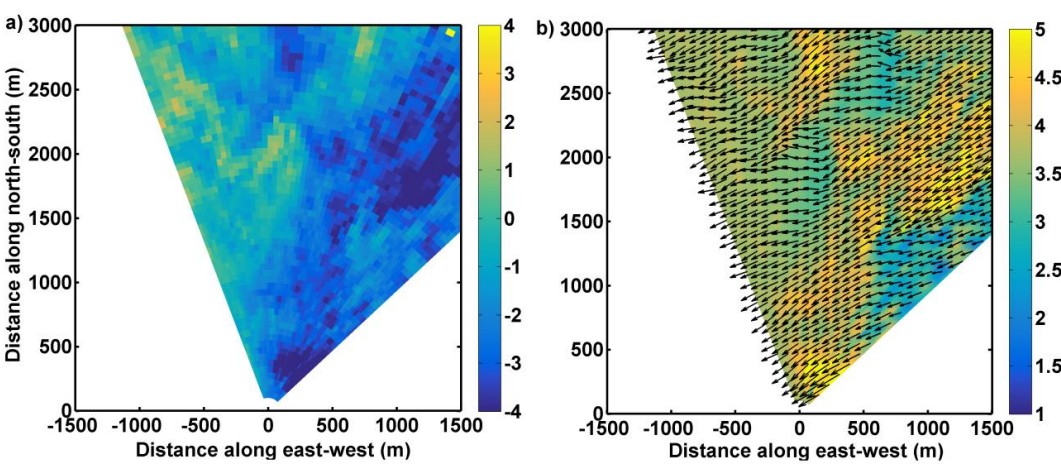




Figure 5. OI retrieval of the horizontal wind field on a PPI scan performed by D1 on April 25[th],
1652 UTC. (a) LOS velocity field (in m s[-1]) measured by D1 (b) Horizontal wind field retrieved
using the OI technique.  The colors indicate magnitude of horizontal wind speed (in m s[-1]) and
wind direction is indicated by arrows.

The OI technique uses Bayesian statistical technique to find a 2-D wind field most

consistent with the LOS velocity observations from the lidar PPI scans.  The technique starts
with a first guess (referred to as "background") of the wind field which is a single VAD estimate
using all the LOS velocities from the PPI scan.  The final wind field is arrived at by adding an
"analysis increment" to the first guess which is estimated using the background and observation
error covariances (see Choukulkar et al. (2012b) for details).  The OI technique does not make
any assumptions about the flow field (such as homogeneity or isotropy), however, it assumes that
the background error is homogeneous.  The validity of this assumption has been tested through
simulated lidar measurements and was found to be reasonable (Choukulkar, 2013).
**3.  Determining Baseline Uncertainty:**

Uncertainties emerge associated with the LOS velocity measurements made by the

Doppler lidar.  These uncertainties can be categorized into: (1) random error in the estimation of
the velocity and (2) error due to the path integration (range-gating) inherent in pulsed Doppler
lidar measurements.  The random component of the LOS velocity estimate, for D1, found by
linearly fitting the autocovariance from lags 1 through 4 and extrapolating to zeroth lag
(Lenschow et al., 2000) is shown in Figure 6.  Similar values were estimated for all three 200S
lidars (D1, D2 and UTD) and the error was a function of Signal to Noise Ratio (SNR) only.





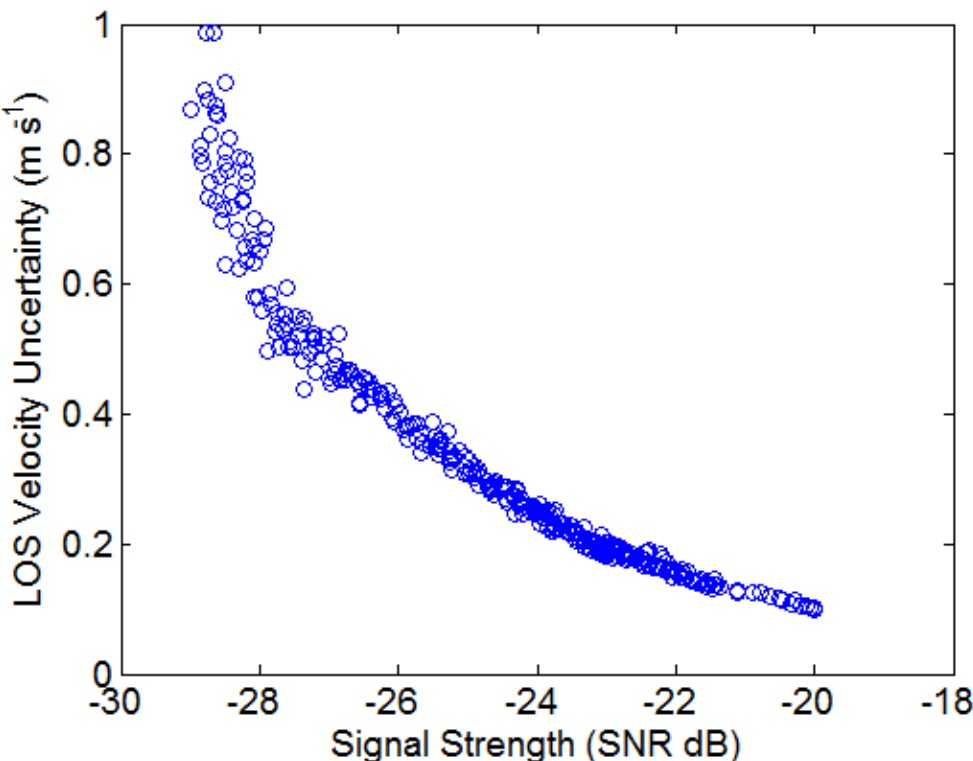


Figure 6.   Estimate of the standard deviation of random error as a function of SNR estimated
using (Lenschow et al., 2000).

In addition to the uncertainty due to the random noise in the LOS velocity estimates, a

systematic underestimation of the variability in the velocity field at shorter length scales is
introduced due to the path averaging of the lidar pulse. To determine the additional uncertainty
due to range-gate averaging, the power spectrum of the lidar LOS velocity measurements was
compared to the power spectrum of the sonic-derived LOS measurements.  Data from a 3-day
period where the 200S lidars (D1, D2 and UTD) were performing hour long stares at each sonic
anemometer level were used.  The spectra of the lidar LOS velocity measurements from the
various hour long stares at each sonic anemometer level were averaged and compared to
correspondingly averaged sonic-derived LOS measurements (see Figure 7a).  As seen in Figure
7a, the spectra from the lidar LOS measurements (blue line) flattens out for frequencies higher
than ~0.25 Hz indicating the variations due to random noise dominate. Once the variations due to



random noise are subtracted from the spectra, the under-prediction of the variability due to the
pulse averaging is clearly visible and can be estimated (see Figure 7b). This under-estimation of
the variability is defined as the square-root of the difference between the spectra of the sonic
anemometer and the lidar measurement and is found to be 0.23 m s$^{-1}$. The under-estimation of
the variability can be interpreted as a smoothing of the lidar measurements and hence adds to the
differences between the lidar and sonic anemometer wind measurements.

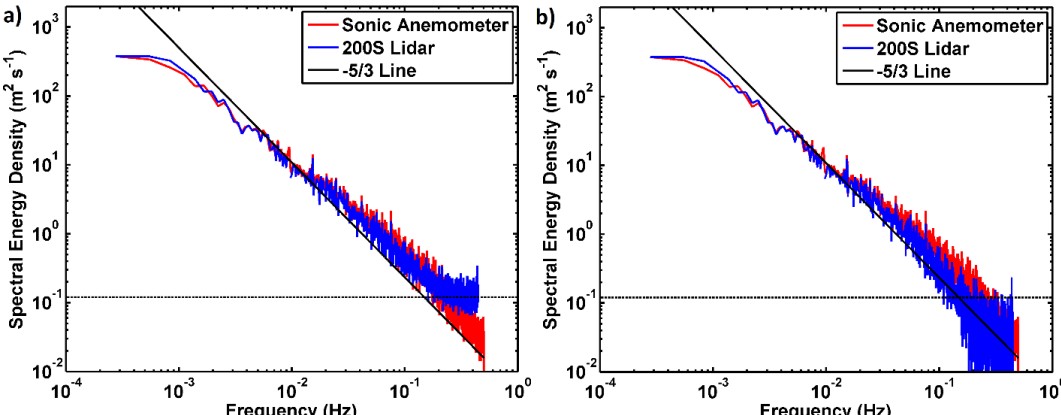


Figure 7.   Comparison of the FFT of LOS data measured by the lidar and the LOS derived from
the sonic anemometer measurements. The solid black line indicates the energy cascade following
the -5/3 Kolmogorov energy spectrum (a) Comparison of the FFT showing the noise floor of the
Doppler lidar measurements (dotted black line). (b) Comparison of the FFT after subtracting the
noise from the lidar FFT. The under-prediction of the variability due to the pulse averaging can
be seen clearly and found to be 0.23 m s$^{-1}$.

Finally, the total difference between the 1 Hz lidar-measured-LOS velocity and the 1 Hz

sonic-derived-LOS velocity measurements, as estimated from direct comparison is presented in

Table 4 (Lundquist et al., 2016a). This difference is slightly larger than the combined

uncertainties from random noise and pulse averaging. This difference in the uncertainty
estimates could be due to some factors that are as yet unaccounted or due to improper estimate of
the uncertainties due to random noise and pulse averaging. The uncertainty of the LOS
measurement by the lidar (when compared to the sonic anemometer) allows evaluating the
various measurement techniques in terms of additional uncertainty added to this baseline value.



The offset in the LOS velocity of the UTD lidar was found to be due to improper calibration of
the pulse length dependent frequency offset (Lundquist et al., 2016b). This was characterized
using independent measurements and was found to be constant throughout the XPIA campaign.
Therefore in all measurements presented in this paper, this static offset has been subtracted from
the UTD lidar LOS velocity.

Table 4. Comparison of the instantaneous lidar LOS measurement to sonic derived LOS
measurement at all sonic levels.

| Lidar | Corr Coef. | Slope | Offset | Std. Dev. of differences |
|---|---|---|---|---|
| D1 | 0.99 | 1.01 | 0.02 m s$^{-1}$ | 0.50 m s$^{-1}$ |
| D2 | 0.98 | 0.98 | -0.12 m s$^{-1}$ | 0.66 m s$^{-1}$ |
| UTD | 0.99 | 1.00 | -0.60 m s$^{-1}$ | 0.53 m s$^{-1}$ |


**4.   Validation of Wind-field Measurements:**

The wind field measurements from the measurement techniques outlined earlier are now

evaluated using the sonic anemometer as the standard. A three-step 6-sigma outlier rejection is
applied in each of the comparisons before estimating the validation metrics. The validation
metrics used here are the mean and standard deviation of the differences between the lidar and
sonic anemometer measurements.
*4.1 Virtual Tower Stares:*

The 3-dimensional wind was measured using the VTS technique by taking 5-s of LOS

velocity data from the three 200S lidars which overlapped in time and space (as defined by the
common volume), and least-squares fitted to Eq. 1 to derive the 3-D wind field. The comparisons
of the 3-dimensional wind field as measured by the VTS technique to 5-s averaged sonic
anemometer measurements is shown in Table 5. These measurements agree with a high
correlation coefficient (0.97 and 0.99 for wind speed and direction respectively) and low
standard deviation of the differences (0.51 m s$^{-1}$ and 10.16° for horizontal wind speed and
horizontal wind direction respectively) between the sonic anemometer and the VTS
measurements. In addition, the vertical velocity measurements also show a reasonably good
correlation coefficient (0.86) and low standard deviation of differences (0.5 m s$^{-1}$). Note that in



the vertical velocity comparisons, only measurements at and above the 150 m sonic are
compared. This is due to the fact that at the lower sonic levels, the elevation angles in the VTS
scans were quite low and as a result the component we are trying to estimate is perpendicular to
the lidar look direction resulting in a noisy vertical velocity retrieval. The velocity retrievals at
the 50 m level from the VTS scans are shown in Figure 8. As can be observed from Figure 8,
there is no skill in the vertical velocity retrievals at low elevation angles.
Table 5. Statistics from comparison of wind field measurements from the VTS to the sonic
anemometer measurements

| Wind Field | Corr Coef | Slope | Offset | Std Dev of Differences |
|---|---|---|---|---|
| Horizontal Wind Speed (all heights) | 0.97 | 0.96 | 0.21 m s$^{-1}$ | 0.50 m s$^{-1}$ |
| Horizontal Wind Direction (all heights) | 0.99 | 0.97 | 3.36º | 9.87º |
| Vertical Velocity (150 m to 300 m) | 0.86 | 1.06 | -0.02 m s$^{-1}$ | 0.50 m s$^{-1}$ |



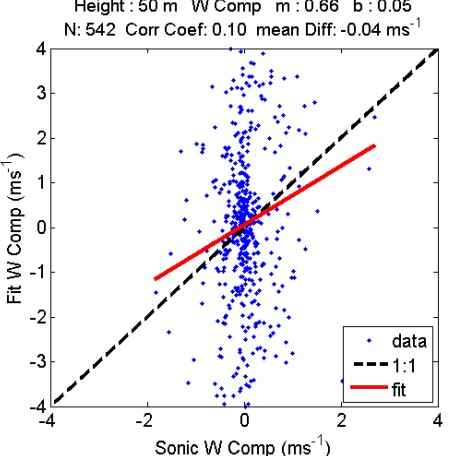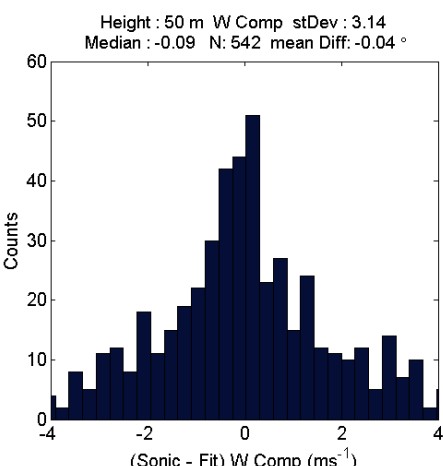


Figure 8. Comparison of the vertical velocity measurements from VTS at the 50 m level with
the sonic anemometer measurements.


### 4.2 Coordinated Sparse Sampling:

The comparison of the SCB measurement point over the BAO tower with 15-s averaged sonic anemometer measurements at the 100 m level are shown in Figure 9. The measurements made from the SCB technique show good agreement with sonic anemometer measurements (as shown in Figure 9) with correlation coefficient of 0.98 and 0.99 for wind speed and wind direction respectively. The correlation coefficient for the vertical velocity was lower (0.54) due to the fact that these measurements were made at the 100 m level which leads to lower skill in vertical velocity measurements as explained in the previous section. The main difference between the VTS measurement strategy and the SCB strategy is the amount of time buffer allowed to make overlapping LOS measurements. In the VTS strategy, each lidar performed a stare scan for 25-s at each measurement location, while it was 5-s for the SCB strategy. Therefore, there was less time to ensure measurement overlap in the SCB strategy and it is expected to have slightly higher uncertainty in wind field measurement.






Figure 9. Comparison of the (a, b) wind speed, (c, d) wind direction and (e, f) vertical velocity

measurements from the SCB technique with the measurements made by the sonic anemometer.





### 4.3 Uncoordinated Virtual Towers (UVT):

The measurements made from the uncoordinated virtual tower (UVT) technique were
compared to the 15-s averaged sonic anemometer measurements at all 6 levels of the BAO
tower. Figure 10 shows that the measurements from the UVT technique have good agreement
with the sonic anemometer measurements with correlation coefficients of 0.95 and 0.99 for wind
speed and wind direction respectively. The standard deviation of the differences (0.65 m s$^{-1}$ for
horizontal wind speed and 11.62$^{\circ}$ for the horizontal wind direction) were found to be slightly
higher compared to the VTS technique. This increase is expected as the LOS velocity
measurements are no longer coordinated in time which leads to an increase in measurement
uncertainty due to non-stationarity of the atmosphere. The vertical velocity measurements made
using this technique for heights 150m and above show a similar skill as the VTS technique,
albeit with a slightly lower correlation coefficient of 0.77. In addition, comparison of the vertical
velocity measurements made at the 50m level (see Figure 11) show that the UVT technique has
no skill in making accurate measurements due to the lower elevation angles involved.







Figure 10. Comparison of the (a, b) wind speed, (c, d) wind direction and (e, f) vertical velocity
measurements from the UVT technique with the measurements made by the sonic anemometer.





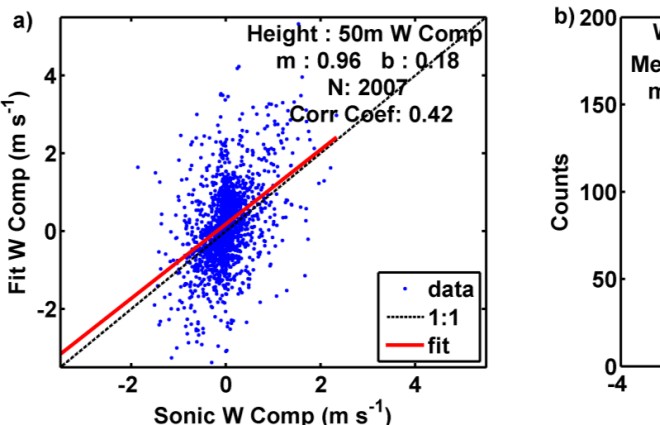

Figure 11. Comparison of the vertical velocity measurements from UVT at the 50 m level with the sonic anemometer measurements.

*4.4 Uncoordinated Volume Scan:*

The uncoordinated volume scan strategy further relaxes the requirement for each lidar to make simultaneous LOS measurements at each measurement location and instead uses all LOS velocity measurements from all Doppler lidars that fall within the grid volume and are within a given time-window (in this case 5-min) to make a wind field measurement. Comparison of the uncoordinated volume scan measurements with 5-min averaged sonic anemometer measurements (at three levels from 50 m to 150 m), show good correlation coefficients of 0.95 and 0.99 for wind speed and direction respectively (see Figure 12).





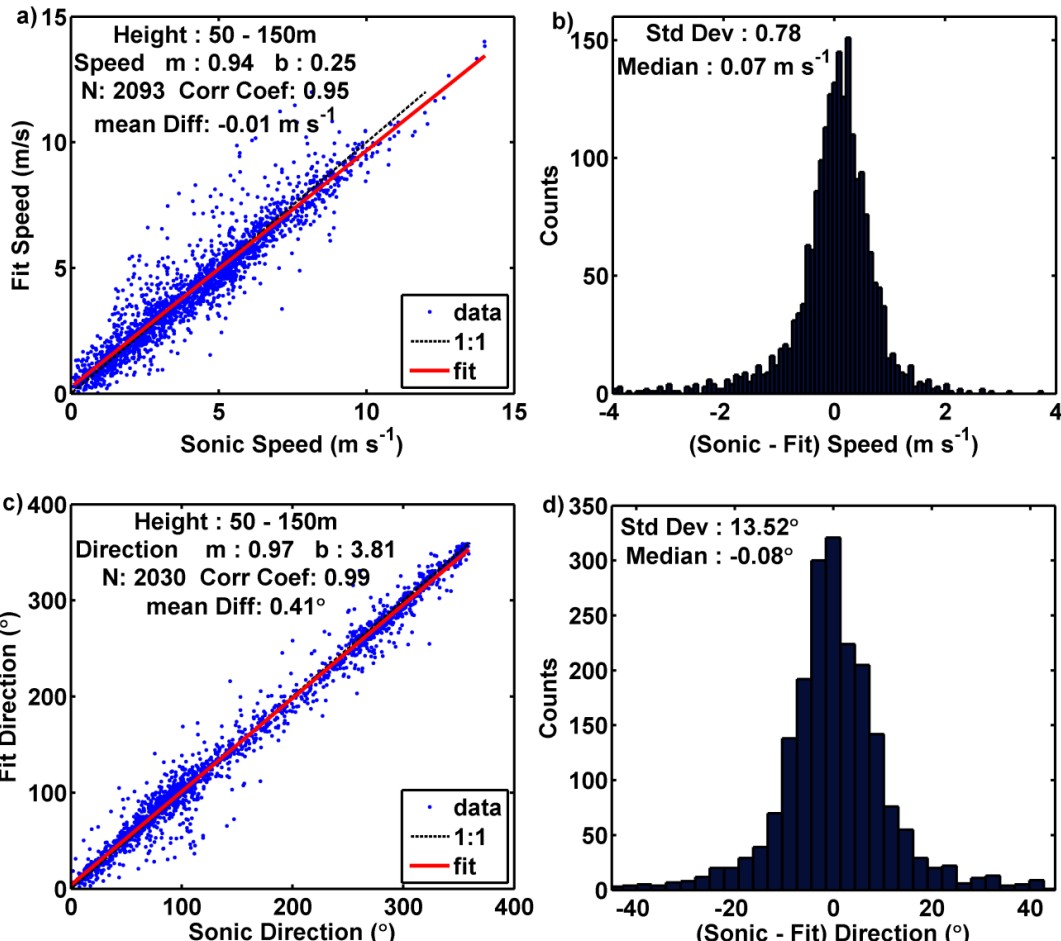

Figure 12. Comparison of the (a, b) wind speed, (c, d) wind direction measurements from the
uncoordinated volume scan technique with the measurements made by the sonic anemometer.

The standard deviation of the differences between the uncoordinated volume scan
measurements and sonic anemometer measurements (0.78 m s$^{-1}$ for horizontal wind speed and
13.52° for horizontal wind direction) are higher compared to the differences reported for the
coordinated measurement techniques. The higher uncertainties could be due to non-stationarity
of the winds over the measurement accumulation period whose effect is expected to be much
larger compared to the case of UVT technique due to the longer measurement accumulation
period. Another factor (which is related to the non-stationarity of the wind) is the less





representative LOS velocity statistics which is due to the fact that since each lidar does not spend
enough time measuring within each grid cell.  As a result, the mean of the LOS velocity
measurements from each lidar are not representative of the mean velocity over which the wind
retrieval is made.
*4.5 Single Doppler Optimal Interpolation (OI) Technique:*

The OI technique allows retrieval of 2-D wind field over conical scans without applying

the assumption of horizontal homogeneity of the wind.  The OI technique was applied to the
sector scans performed by each lidar in the uncoordinated volume scan technique.  Each sector
scan took 30-s to complete, and hence the OI retrieval is compared to 30-s averaged sonic
anemometer measurement shown in Figure 13. The retrievals from the OI technique agree with
the sonic anemometer measurements quite well with correlation coefficients of 0.93 and 0.98 for
wind speed and wind direction respectively.  The standard deviation of the differences (1.04 m s$^{-1}$
$^{1}$ for horizontal wind speed and 20.74$^{\circ}$ for horizontal wind direction) are higher, compared to the
uncoordinated volume scan technique.






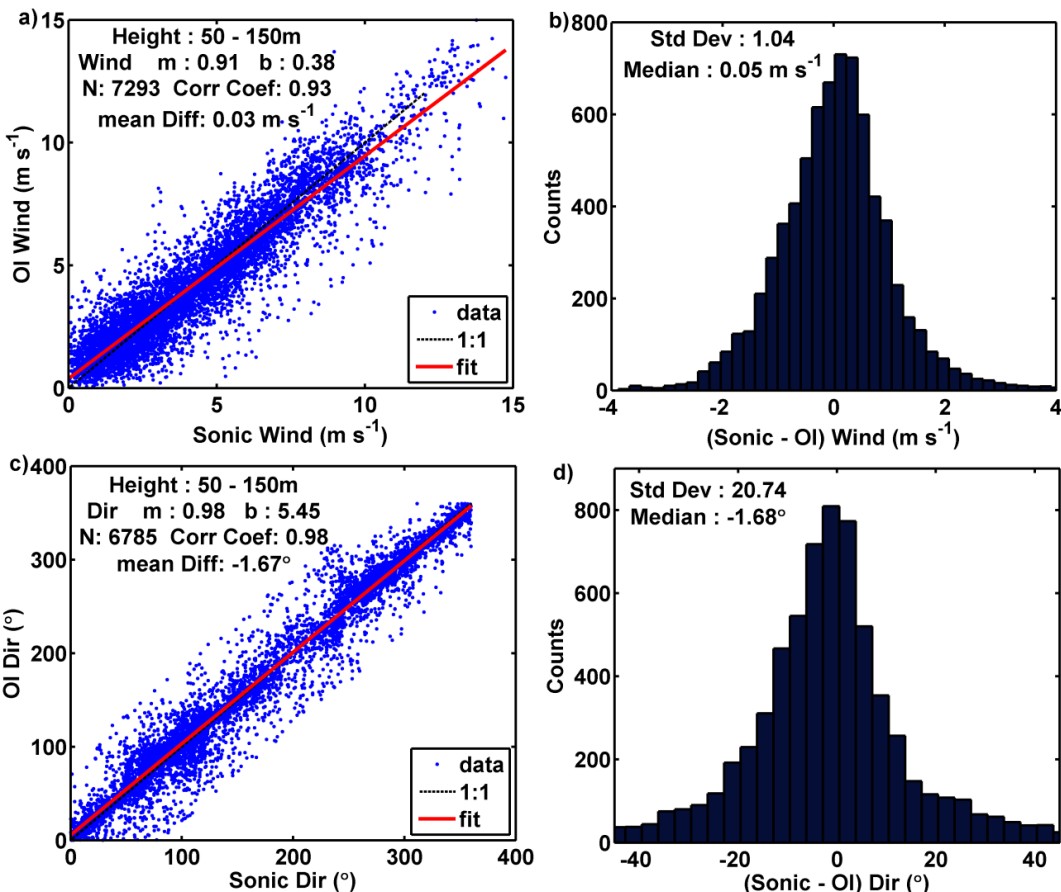


Figure 13. Comparison of the (a, b) wind speed, (c, d) wind direction measurements from the
single Doppler OI technique with the measurements made by the sonic anemometer.

**5. Discussion of Results**

The precision of the wind measurement (defined as the standard deviation of the
differences between the Doppler measurement and the sonic anemometer measurement) obtained
from the various Doppler lidar techniques can now be compared. The measurement precision
reported for each of the techniques is related to time it takes to perform one measurement, and
hence the time average used to evaluate the technique. This method is chosen so that the
inherent trade-off between spatial coverage and temporal resolution is clear.



Figure 14 shows the comparison of the uncertainties for the LOS velocity as well as the
estimates of the horizontal wind speed and direction from the various measurement techniques.
As seen from Figure 14, the uncertainty in the LOS velocity (from comparison with sonic
anemometer) is 0.5 m s$^{-1}$. It is observed that with an averaging time of 5-s for the VTS method,
the uncertainty does not increase compared to the sonic-LOS velocity uncertainty. The most
probable reason the precision of the VTS technique is found to be the highest (compared to other
velocity retrieval techniques presented here) is due to the fact that the three 200S lidars made
simultaneous measurements within the common volume and thus relying less heavily on the
assumption of stationarity of the atmosphere or spatial homogeneity.  Increasing the
measurement complexity and spatial coverage either through faster scanning or relaxing the
requirement of temporal coordination, the measurement uncertainty increases as well.  While the
single Doppler OI does not require complex scanning technique, it requires certain assumptions
as part of the retrieval process (Choukulkar et al., 2012) and hence increases the measurement
uncertainty.




Figure 14. Measurement uncertainties for (a) horizontal wind speed and (b) horizontal wind
direction estimated for the different measurement strategies investigated.





*Effect of Stability:*


The precision of wind measurements is also evaluated in various stability conditions.
The stability is defined using hourly averaged virtual potential temperature gradient between the
50 m level and the 300 m level using both the tower measurements and radiometer
measurements (Bianco et al., 2016). Conditions were determined to be stable for positive
gradient of the virtual potential temperature and unstable for a negative gradient of the virtual
potential temperature.
The lidar wind speed measurements are found to be slightly more precise during stable
conditions, compared to unstable conditions (see Figure 15). The higher uncertainties observed
during unstable conditions might be due to the fact that unstable conditions show higher
variability than stable conditions which might lead to higher level of uncertainty. However, no
consistent pattern emerges for the effect of stability on wind direction uncertainty. This might be
due to the fact that all the stable conditions examined here were accompanied by low wind
conditions which usually leads to higher variability in wind direction while the unstable
conditions had higher wind speeds. As a result, the wind direction uncertainty during stable
conditions is found to be higher. It is clear from Figure 15 that stability and spatial variability do
have a significant impact on the measurement uncertainty.





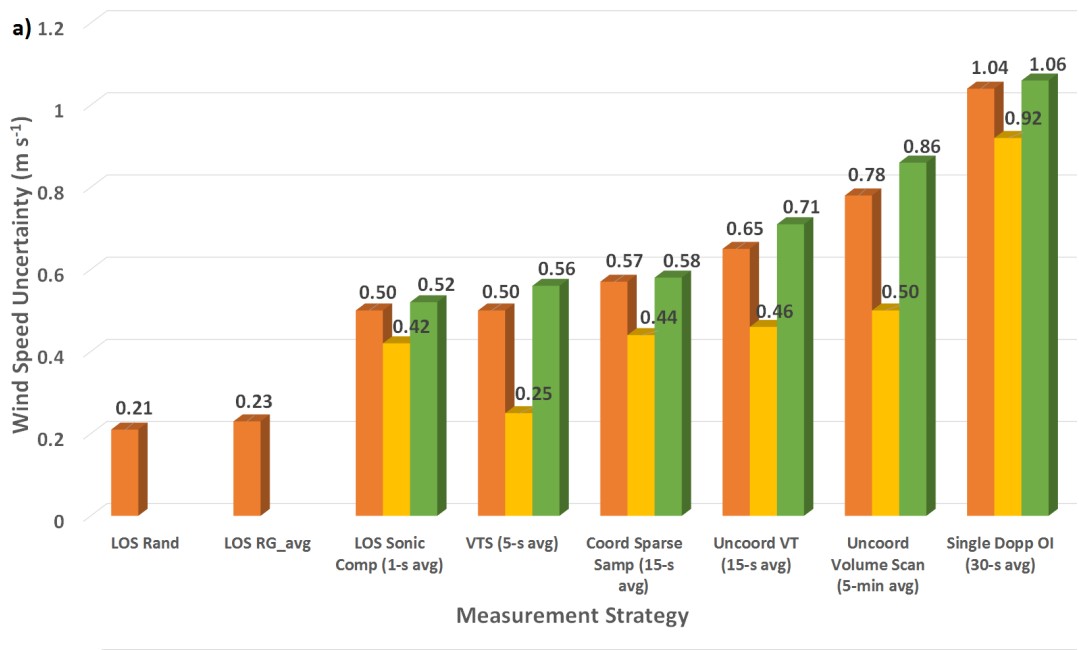

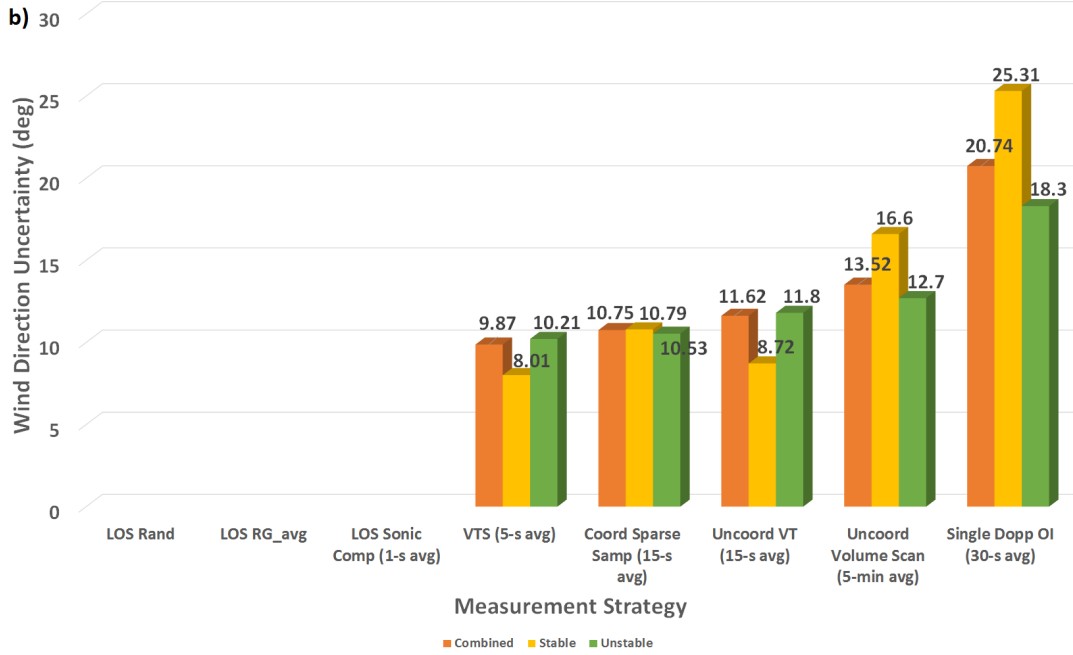


Figure 15. Measurement uncertainty as a function of stability for (a) horizontal wind speed and
(b) horizontal wind direction.





## 6. Conclusions


Scanning Doppler lidars are powerful tools that enable measuring atmospheric flows
using various configuration of single and multi-Doppler techniques. An important aspect of
proper interpretation of the measurements made using Doppler lidars is understanding the
inherent uncertainties associated with the corresponding measurement technique. In this paper,
the uncertainties associated with Doppler lidar measurements were quantified starting with the
uncertainties due to random noise and pulse averaging to uncertainties associated with single and
multi-Doppler measurement techniques.
It was found that as complexity of the measurement technique and/or the spatial coverage
of measurements made increased, the uncertainty in the wind measurement also increased. For
multi-Doppler measurements, the magnitude of uncertainty was associated with ability to make
coordinated measurements. Measurements made using accurate coordinated scanning resulted in
lower uncertainties compared to measurements from temporally uncoordinated scanning. This
result is expected due to the non-stationarity of the atmosphere and presence of spatial variability
in the wind field. The single Doppler OI technique resulted in the highest measurement
uncertainty (compared to multi-Doppler techniques), but also had the largest spatial coverage at
high update rates and is less expensive as a result of requiring only one lidar.
The results illustrate the trade-off between making highly precise measurements at one
location versus accepting a lower precision but covering larger spatial extents. Although the
magnitude of the uncertainties for the various measurement techniques presented in this paper
might not be reproducible at other locations and under different wind conditions, the trends
observed should be similar. This quantification of the uncertainty as a function of measurement
technique allows proper selection of measurement strategy given the goals of the experiment and
interpretation of the measurements made using those techniques.

## 7. Acknowledgements


The authors acknowledge the funding for this work provided by the U.S. Department of
Energy, Office of Energy Efficiency and Renewable Energy and by NOAA's Earth System
Research Laboratory. The authors also acknowledge contributions of numerous individuals and
organization who assisted with the field deployment including Bruce Bartram, Duane Hazen,
Tom Ayers, Jesse Leach, Paul Johnston, Lefthand Water District, Erie High School and the St.





Vrain School District.  We also express our appreciation to NOAA/Earth Systems Research

Laboratory/Physical Sciences Division for supporting the instrumentation at the BAO facility.

We express appreciation to the National Science Foundation for supporting the CABL

deployments (https://www.eol.ucar.edu/field_projects/cabl) of the tower instrumentation. NREL

is a national laboratory of the U. S. Department of Energy, Office of Energy Efficiency and

Renewable Energy, operated by the Alliance for Sustainable Energy, LLC.

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
