# Peer review of "Evaluation of Single- and Multiple-Doppler Lidar Techniques to Measure"

_Atmospheric Measurement Techniques, 2016_

## Referee Comment (RC1) · Anonymous Referee #1 · 11 Nov 2016

Referee comments:

**An initial paragraph or section evaluating the overall quality of the discussion paper ("general comments"):**

The paper presents an overview of the comprehensive wind lidar intercomparison field test XPIA conducted around the 300 m BOA met tower in Colorado April 2015.

The paper present and address wind lidar measurement capability and uncertainties with reference to sonic anemometry installed and operated simultaneously the met tower every 50 meter.

In total, five pulsed wind lidar systems were co-operated and their measurement capabilities are described and data retrieval algorithms and data quality including uncertainty with the reference to the Sonics intercom pared.

The paper shows the capability with the state of the art commercial available Leosphere scanners 200S and compare them with scientific grade pulsed when operated in staring mode and in standard PPI scanning mode.

**a section addressing individual scientific questions/issues ("specific comments"),**

The lidar measured wind fields are presented using several data retrieval methodologies, using single lidars with interpolation techniques and also multiple, but all lidar beams and measured line of sight Doppler velocities were unsynchronized and un-coordinated while measuring.

The quality of the resulting wind field measurements seems, in addition to large probe volumes, signal-to-noise issues and laser beam pointing accuracy seems challenged also by select tradeoff between relative low sampling rate and the quest to cover large spatial coverage.

The logistical setup of the lidars around the BOA tower seems not to be ideal but impaired by the BOA towers surroundings land use, resulting in that the experimental setup, including angles of vertical inclination and distances were not ideal for demonstrating the full potential capabilities of the pulsed lidars. Nevertheless, given these constraints, which seems to be imposed by the surroundings, this paper documents to my knowledge the first lidar-to lidar intercomparison study between different groups lidar systems, and as such, it presents a study on capability, including data retrieval methodologies, and uncertainties, that have not previous been described using lidars.

As such although not ideal, this paper should be published as one of the first references for multiple joint lidar wind measurement capability and intercomparison.

The lidars applied all seem to be limited in their scanning operation to perform either PPI or RHI scanning. This is unfortunate and limits the best use of the lidars. Unfortunately, at the time of XPIA, limited scan mode represented the state of the art. However, if user defined multiple lidar coordinated trajectory scanning had been available the trajectory scanning over large volumes of air could be more flexible and efficient.

**Listing of purely technical corrections**

It is suggested to amend the references in the paper with previous and recent publication about other staring lidar field tests with the following publications:

Line 95: I suggest to amend the ref list re triple lidars by adding a ref to Simley et al[1]

Line 96: regarding long-range triple scanning lidars, add ref to Vasiljevic et al. 2016 [2]

Line 120: regarding triple staring lidars, add ref to Mann et al. 2009 [3]

Line 159: the ref is not found in the ref list?

Line 161: foot note: need to explain why this particular lidar needed additional accumulation time- it's the lidar closest to the BAO tower hence it should have the highest CNR?

Line 177 for helping the reader it could be stated that this corresponds to approximately +/- 1 meter pointing uncertainty at the 1 km range

Line 191: add a ref to Mann et al. (2009) again following the ref to Calhoun 2006

Line 311: 2 the meaning of the statement "from lags 1 through 4" is not clear to me, is 1-4 range gates, if so pls specify?

Line 320: Stating: "To determine the additional uncertainty due to range averaging" The word "uncertainty" may not be the most correct word to use here, the lidar's range-average results in a deterministic filtering effect, that filters variance at short length scales, but this is not an uncertainty issue, rather an instrument filtering effect, which is common to all probe volume averaging instruments, lidars included. Could it be stated as: To quantify the variance reduction due to spatial averaging of the lidars Line-of-sight probe volume, …

[1]     Simley E, Angelou N, Mikkelsen T, Sjöholm M, Mann J and Pao L Y 2016 Characterization of wind velocities in the upstream induction zone of a wind turbine using scanning continuous-wave lidars *J. Renew. Sustain. Energy* **8** 13301

[2]     Vasiljevi N, Lea G, Courtney M, Cariou J, Mann J and Mikkelsen T 2016 Long-range WindScanner system *Remote Sens.* **Dec 2016** 1–24

[3]     Mann J, Cariou J-P, Courtney M S, Parmentier R, Mikkelsen T, Wagner R, Lindelöw P, Sjöholm M and Enevoldsen K 2009 Comparison of 3D turbulence measurements using three staring wind lidars and a sonic anemometer *Meteorol. Zeitschrift* **18** 135–40

---

## Referee Comment (RC2) · Anonymous Referee #2 · 5 Dec 2016

The paper "Evaluation of Single- and Multiple-Doppler Lidar Techniques to Measure Complex Flow during the XPIA Field Campaign" presents an overview of single- and multi- Doppler scanning methods operated during the XPIA campaign. The focus is on the scanning method description and their ability to retrieve the wind vector components. The described scanning methods consist of PPI and RHI scans, and thus the methods themselves are not novel. From the manuscript, readers can conclude that retrieval of the wind vector components using unsynchronized multiple scanning lidars brings poor accuracy. This is logical due to the non-stationarity of the atmosphere (the author commented this as well). Therefore, this information does not bring anything new to readers. The author concluded that as the complexity of the scanning

method increased the uncertainty in the wind vector retrieval increased as well. This is a misleading conclusion. In this specific case, the increase in complexity brought the increase in the lag between the lidars, which in turn resulted in a poorer estimation of the wind vector components (as expected). Furthermore, the author presented the comparison between the retrieved single- or multi-Doppler wind vector acquired over the shortest time period (not averaged = 'instantaneous') and the averaged sonic data. I find this approach odd. In the case of multi-Doppler retrievals, lidars were not synchronized and comparing the retrieved information without first averaging over a certain period will not produce good results (common sense). The same stands for the OI method. I suggest to the author to reformulate the paper and investigate trade-offs between the averaging period and spatial coverage instead. Furthermore, I strongly suggested to the author to follow the Vancouver protocol, and include only those co-authors that substantially contributed to: 1) conception and design of the study, or analysis and interpretation of data 2) drafting of the manuscript or revising it critically for important intellectual content 3) the final approval of the version to be published

---

## Author Comment (AC1) · 21 Dec 2016

**Responses to referee comments:**

**An initial paragraph or section evaluating the overall quality of the discussion paper ("general comments"):**

The paper presents an overview of the comprehensive wind lidar intercomparison field test XPIA conducted around the 300 m BOA met tower in Colorado April 2015.

The paper present and address wind lidar measurement capability and uncertainties with reference to sonic anemometry installed and operated simultaneously the met tower every 50 meter.

In total, five pulsed wind lidar systems were co-operated and their measurement capabilities are described and data retrieval algorithms and data quality including uncertainty with the reference to the Sonics intercom pared.

The paper shows the capability with the state of the art commercial available Leosphere scanners 200S and compare them with scientific grade pulsed when operated in staring mode and in standard PPI scanning mode.

The authors thank the referee for taking the time to review this manuscript and providing their valuable inputs.

**a section addressing individual scientific questions/issues ("specific comments"),**

The lidar measured wind fields are presented using several data retrieval methodologies, using single lidars with interpolation techniques and also multiple, but all lidar beams and measured line of sight Doppler velocities were unsynchronized and un-coordinated while measuring.

This manuscript investigates both temporally synchronized and un-synchronized measurement uncertainty.  The authors would like to note that while it is true the lidar measurements discussed here were not coordinated using techniques described in Vasiljevic et al (2016), temporal coordination was achieved.   As summarized in Table 2 and 3, the Doppler lidars were able to maintain pointing uncertainty of less than 0.15° (+/- 2.5m @ 1km range) and time uncertainty of less than 0.4 s.  Since, all our averaging periods are much greater than 0.4 seconds and the common volume is much bigger than the pointing uncertainty, we feel that the lidars were able to make synchronized measurements (in time and space) when we aimed to do so.  We have renamed the sections to make it clearer where synchronized scanning was used and where the measurements were allowed to be unsynchronized.

The quality of the resulting wind field measurements seems, in addition to large probe volumes, signal-to noise issues and laser beam pointing accuracy seems challenged also by select tradeoff between relative low sampling rate and the quest to cover large spatial coverage.

The measurement techniques investigated span the range of highest sampling frequency with lowest spatial coverage (virtual tower stare scans) and highest possible spatial coverage but lower sampling frequency (multi-Doppler volume scans) due to mechanical and data rate limitations.

The logistical setup of the lidars around the BOA tower seems not to be ideal but impaired by the BOA towers surroundings land use, resulting in that the experimental setup, including angles of vertical inclination and distances were not ideal for demonstrating the full potential capabilities of the pulsed lidars. Nevertheless, given these constraints, which seems to be imposed by the surroundings, this paper documents to my knowledge the first lidar-to lidar intercomparison study between different groups lidar systems, and as such, it presents a study on capability, including data retrieval methodologies, and uncertainties, that have not previous been described using lidars.

The authors agree that the setup is non-ideal.  However, even though two of the lidars have close to 180 difference in azimuth, they have substantially different elevation angles when interrogating volumes near the BAO tower and hence do provide some unique velocity information.

As such although not ideal, this paper should be published as one of the first references for multiple joint lidar wind measurement capability and intercomparison.

The lidars applied all seem to be limited in their scanning operation to perform either PPI or RHI scanning. This is unfortunate and limits the best use of the lidars. Unfortunately, at the time of XPIA, limited scan mode represented the state of the art. However, if user defined multiple lidar coordinated trajectory scanning had been available the trajectory scanning over large volumes of air could be more flexible and efficient.

The authors agree with the reviewer about this as the operation of the lidars was limited by the GUI provided by the manufacturer.  In addition, as many commercially available lidar systems currently are not capable of performing more complicated scanning trajectories, this paper discusses the possibilities available to such lidar systems.  We have added text to the manuscript to make this point.

**Listing of purely technical corrections**

It is suggested to amend the references in the paper with previous and recent publication about other staring lidar field tests with the following publications:

Line 95: I suggest to amend the ref list re triple lidars by adding a ref to Simley et al[1]

Added

Line 96: regarding long-range triple scanning lidars, add ref to Vasiljevic et al. 2016 [2]

Added

Line 120: regarding triple staring lidars, add ref to Mann et al. 2009 [3]

Added

Line 159: the ref is not found in the ref list?

Fixed

Line 161: foot note: need to explain why this particular lidar needed additional accumulation time- it's the lidar closest to the BAO tower hence it should have the highest CNR?

Added: "… to ensure coverage during multi-Doppler volume scans"

Line 177 for helping the reader it could be stated that this corresponds to approximately +/- 1 meter pointing uncertainty at the 1 km range

Added

Line 191: add a ref to Mann et al. (2009) again following the ref to Calhoun 2006

Added

Line 311: 2 the meaning of the statement "from lags 1 through 4" is not clear to me, is 1-4 range gates, if so pls specify?

The lags refers to the lags of the autocovariance of the LOS velocity. Modified the statement to make this clearer.

Line 320: Stating: "To determine the additional uncertainty due to range averaging" The word "uncertainty" may not be the most correct word to use here, the lidar's range-average results in a deterministic filtering effect, that filters variance at short length scales, but this is not an uncertainty issue, rather an instrument filtering effect, which is common to all probe volume averaging instruments, lidars included. Could it be stated as: To quantify the variance reduction due to spatial averaging of the lidars Line-of-sight probe volume, …

We agree with the reviewer that the effect of the range-gate volume is to filter the variations at smaller scales. However, this can result in differences in the estimate of the LOS velocity (and

its derivatives) when comparing with a "point" measurement such as a sonic anemometer. Hence, in this context we refer to the effect of the volume averaging as contributing to the uncertainty.

[1] Simley E, Angelou N, Mikkelsen T, Sjöholm M, Mann J and Pao L Y 2016 Characterization of wind velocities in the upstream induction zone of a wind turbine using scanning continuous-wave lidars *J. Renew. Sustain. Energy* **8** 13301

[2] Vasiljevi N, Lea G, Courtney M, Cariou J, Mann J and Mikkelsen T 2016 Long-range WindScanner system *Remote Sens.* **Dec 2016** 1–24

[3] Mann J, Cariou J-P, Courtney M S, Parmentier R, Mikkelsen T, Wagner R, Lindelöw P, Sjöholm M and Enevoldsen K 2009 Comparison of 3D turbulence measurements using three staring wind lidars and a sonic anemometer *Meteorol. Zeitschrift* **18** 135–40

---

## Author Comment (AC2) · 21 Dec 2016

**Responses to referee comments:**

The paper "Evaluation of Single- and Multiple-Doppler Lidar Techniques to Measure Complex Flow during the XPIA Field Campaign" presents an overview of single- and multi- Doppler scanning methods operated during the XPIA campaign. The focus is on the scanning method description and their ability to retrieve the wind vector components.

The authors thank the reviewer for taking the time to review this manuscript and providing their valuable inputs.

The described scanning methods consist of PPI and RHI scans, and thus the methods themselves are not novel.

The authors agree with the reviewer's opinion that the scanning strategies themselves are well known and have been discussed quite extensively in literature (many of which have been summarized and cited in the manuscript). In addition, the operation of the lidars was limited by the GUI provided by the manufacturer which allowed programming only simple scan geometries. The goal of this paper is to evaluate each of these techniques against one another and understand the trade-offs in terms of temporal resolution and spatial coverage and understanding flow heterogeneity (to our knowledge, this manuscript seems to be the first to do so). In addition, as many commercially available lidar systems currently are not capable of performing more complicated scanning trajectories, this paper discusses the possibilities available to such lidar systems.

From the manuscript, readers can conclude that retrieval of the wind vector components using unsynchronized multiple scanning lidars brings poor accuracy. This is logical due to the non-stationarity of the atmosphere (the author commented this as well). Therefore, this information does not bring anything new to readers.

In this paper the measurement precision of several temporally synchronized and temporally unsynchronized measurement strategies is discussed. To the authors' knowledge, these techniques have not been compared against one another using a common wind measurement standard. In addition, while the result that unsynchronized scanning will result in higher uncertainty is logical, the degradation has not been quantified. In addition, the degradation is not just due to the non-stationarity of the atmosphere, but also sampling errors introduced as a result of the scanning strategy. Apart from quantifying degradation in measurement precision due to unsynchronized scanning, this paper dwells into quantifying the instrument random error as well as uncertainty due to volume averaging of the laser pulse. The authors feel these aspects do bring new information to the readers and informs on the magnitudes of these errors with respect to differences observed in lidar versus sonic anemometer comparisons.

The author concluded that as the complexity of the scanning method increased the uncertainty in the wind vector retrieval increased as well. This is a misleading conclusion. In this specific

case, the increase in complexity brought the increase in the lag between the lidars, which in turn resulted in a poorer estimation of the wind vector components (as expected).

The authors point to two possible sources (which depend on scanning technique) that add uncertainty to the wind measurement: (1) lack of temporal synchronization (or lag) in the measurements and (2) less-representative LOS velocity statistics.  As the reviewer points out more complex scan strategies will result in higher lag.  However, they can also result in less representatives LOS velocity statistics as there is not enough time spent in each measurement volume due to higher scan speeds and covering larger spatial extents.  The authors refer to the combined effect of these two sources of error when referring to increase in complexity causes higher uncertainty.  We have added further statements in the discussion to ensure this point is comes across clearly.

Furthermore, the author presented the comparison between the retrieved single- or multi-Doppler wind vector acquired over the shortest time period (not averaged = 'instantaneous') and the averaged sonic data. I find this approach odd.

The wind measurement from the various multi-Doppler are estimated by chi-squared fitting the radial velocity equation to the LOS velocity measurements.  Therefore, when 5-seconds of LOS velocity measurements (for example) from various Doppler lidars are fitted to the radial velocity equation, we get an estimate of the 5-s mean velocity as represented by the LOS velocities from the different lidars.  This estimate of the 5-s mean is then compared to the 5-s mean from the sonic anemometer measurements.  Therefore, equally temporally averaged quantities from the lidars and sonic anemometers are compared.

The reason for using the measurement period to determine precision was to ensure each technique was evaluated fairly.  For example, in the case of the virtual tower stares, each wind measurement is achieved every 5 seconds, while for the multi-Doppler volume scan it takes 5 minutes (but it covers a larger spatial area).  If measurements were averaged to a common time period (say 10 minutes) there would be many more samples from the virtual tower stares compared to the multi-Doppler volume scan and hence, it will not be an equal comparison.

In the case of multi-Doppler retrievals, lidars were not synchronized and comparing the retrieved information without first averaging over a certain period will not produce good results (common sense). The same stands for the OI method. I suggest to the author to reformulate the paper and investigate trade-offs between the averaging period and spatial coverage instead.

This manuscript investigates both temporally synchronized and un-synchronized measurement uncertainty.  The authors would like to note that while it is true the lidar measurements discussed here were not coordinated using techniques described in Vasiljevic et al (2016), temporal synchronization was achieved.   As summarized in Table 2 and 3, the Doppler lidars

were able to maintain pointing uncertainty of less than 0.15° (+/- 2.5m @ 1km range) and time uncertainty of less than 0.4 s.  Since, all our averaging periods are much greater than 0.4 seconds and the common volume is much bigger than the pointing uncertainty, we feel that the lidars were able to make synchronized measurements (in time and space) when we aimed to do so.  We have renamed the sections to make it clearer where synchronized scanning was used and where the measurements were allowed to be unsynchronized.

In the case of the OI method, a similar reasoning as explained in our previous response is used for evaluating measurement accuracy.  The OI method uses all the LOS velocity measurements from a sector scan to estimate the 2-D wind field on the sector scan.  Therefore, the OI results represents an estimate of the wind over the time it took to perform the scan (in this case 30 seconds).

Furthermore, I strongly suggested to the author to follow the Vancouver protocol, and include only those coauthors that substantially contributed to: 1) conception and design of the study, or analysis and interpretation of data 2) drafting of the manuscript or revising it critically for important intellectual content 3) the final approval of the version to be published

The XPIA field campaign was a multi-institutional effort involving deployment of a vast array of sensor, both in-situ and remote sensing.  This paper was made possible through a close collaboration among all the persons involved in experiment design, field deployment, data collection and analysis.  Therefore, I feel each of them has earned the authorship.
The reasoning behind including the authors in this manuscript is given below:

Julie Lundquist and James Wilczak – PIs on the proposal and responsible for conception and design of the field study

Alan Brewer, Michael Hardesty, Timothy Bonin, Valerio Iungo, Mithu Debnath, Laura Bianco Aditya Choukulkar, J. Lundquist and J. Wilczak – experiment design, data analysis, manuscript preparation and critical review

Scott Sandberg and Ann Weickmann (NOAA lidars), Ryan Ashton (UTD lidar), Ruben Delgado (UMBC lidar), Steven Oncley and Daniel Wolfe (BAO tower and sonic anemometers) – instrument deployment, instrument operation and maintenance, data collection and initial processing.